# Potential Causal Association Between Atrial Fibrillation/Flutter and Primary Open-Angle Glaucoma: A Two-Sample Mendelian Randomisation Study

**DOI:** 10.3390/jcm13247670

**Published:** 2024-12-16

**Authors:** Young Lee, Je Hyun Seo

**Affiliations:** Veterans Medical Research Institute, Veterans Health Service Medical Center, Seoul 05368, Republic of Korea; lyou7688@gmail.com

**Keywords:** primary open-angle glaucoma, mendelian randomisation, atrial fibrillation, atrial flutter, sleep apnoea, single-nucleotide polymorphisms

## Abstract

**Background**: A few studies have reported controversial relationships between atrial fibrillation/flutter (AF/L) and primary open-angle glaucoma (POAG). This study aimed to investigate the potential causal relationship between AF/L and POAG. **Methods**: Single-nucleotide polymorphisms associated with exposure to AF/L were selected as instrumental variables with significance (*p* < 5.0 × $10^{-8}$) from a genome-wide association study (GWAS) by FinnGen. The GWAS summary of POAG from the UK Biobank was used as the outcome dataset. A two-sample Mendelian randomisation (MR) study was performed to assess the causal effects of AF/L on POAG. In addition, potential confounders, including hypertension, autoimmune hyperthyroidism, sleep apnoea, and alcohol use disorder, were assessed using multivariable MR analysis. **Results:** There was a significant causal association of AF/L with POAG (odds ratio [OR] = 1.26, 95% confidence interval [CI] = 1.07–1.48, *p* = 0.005 using inverse-variance weighting [IVW]). Multivariable MR analysis confirmed a causal association of AF/L with POAG (OR = 1.24, 95% CI = 1.02–1.51, *p* = 0.034 using IVW), but hypertension, hyperthyroidism, sleep apnoea and alcohol use disorder did not show significant causal associations with POAG (all *p* > 0.05). **Conclusions:** This established causal relationship between AF/L and POAG supports the need for further investigation into the role of AF/L as a possible risk factor for POAG. Further research is required to confirm these findings.

## 1. Introduction

Primary open-angle glaucoma (POAG), a major cause of permanent vision loss, is a progressive optic neuropathy characterised by the degeneration of retinal ganglion cells and their axons [1]. Its pathogenesis remains unclear, despite the established roles of multiple factors in its pathophysiology. Elevated intraocular pressure (IOP) is a major risk factor for POAG [2], although the precise mechanisms underlying glaucomatous optic neuropathy and the associated visual field loss remain unclear [1]. However, reports indicate that lowering IOP with long-acting drug delivery reduces the risk of glaucoma progression, highlighting that IOP is a critical factor in glaucoma [3]. Studies identifying the causal risk factors for POAG may enable the early detection and prevention of glaucoma; consequently, they form the foundation for research in ophthalmology. Several risk factors, such as ageing, hypoxia, neuroprotection, and environmental factors, have been suggested as contributing to the pathogenesis of POAG [1,4,5,6,7,8,9]. Minor injuries from repeated reperfusion, such as fluctuating IOP or disrupted autoregulation, may eventually cause oxidative stress and glaucomatous damage [10]. Cardiovascular disorders, including vasospasm, hypertension, and hypotension, are also potential risk factors for glaucoma [11].

Atrial fibrillation (AF) and atrial flutter (AFL) are disorders of significant importance to public health, especially when considering their associations with ischaemic stroke and heart failure [12,13]. AF is the most prevalent type of cardiac arrhythmia, caused by aberrant electrical activity within the atria of the heart, leading to fibrillation. AFL is similar to AF, but the rapid upper-chamber cardiac arrhythmias encountered in clinical practice are often more regular [14]. Additionally, many patients with AFL develop AF over time [15]. Recent advances in the understanding of the unique electrophysiological processes underlying AF and AFL have led to anatomy-based treatment approaches. Owing to their mechanistic similarities, AF and AFL (AF/L) are often classified and studied together. AF prevalence ranges from 2% in the general population to 10–12% in those aged ≥80 years. AF/L presents as an erratic heartbeat that may cause instability in ocular perfusion pressure. AF/L plays a significant role in heart failure, stroke, cardiovascular morbidity, and sudden death. However, the associations between AF/L, as a major cardiovascular disease, and ophthalmological diseases have not been intensively examined. A recent analysis from the Ural Eye and Medical Study showed that AF/L was not associated with major ocular diseases, including glaucoma (*p* = 0.90) [16]. However, Flammer et al. suggested that ischaemia and reperfusion damage caused by unstable ocular perfusion may play a role in the vascular pathophysiology of POAG [17]. An observational study also revealed that patients with normal-tension glaucoma had a higher AF incidence than controls (*p* = 0.022) [18]. Another recent longitudinal study evaluating the association of AF and glaucoma using Korean National Health Insurance data determined that AF increased the risk of developing glaucoma (hazard ratio = 1.31; 95% confidence interval [CI] = 1.15–1.48) [19]. Evidently, research on the relationship between AF and glaucoma is not enough.

Mendelian randomisation (MR) is a genetic epidemiological technique that uses genetic variants associated with potential risk factors (exposures) as instrumental variables (IVs) to assess their causal effects on disease outcomes [20,21]. Several studies have used MR to investigate glaucoma and its potential risk factors [22,23,24]. However, no MR studies have assessed the association between AF/L and POAG. Therefore, this study aimed to analyse the association between AF/L and POAG using univariate MR analysis and to conduct multivariate MR analysis to account for potential confounding factors, such as hypertension, hyperthyroidism, sleep apnoea, and alcohol use disorder. To achieve this goal, a two-sample MR technique was used to examine the causal effects of AF/L on POAG, utilising summary statistics from FinnGen [25] for exposures and from the UK Biobank (UKB) for outcomes [26].

## 2. Materials and Methods

### 2.1. Study Design

The study protocol was approved by the Institutional Review Board of the Veterans Health Service Medical Center (approval number. 2023-12-030), and owing to the retrospective study design, the requirement for informed consent was waived. This study was conducted in accordance with the principles of the Declaration of Helsinki.

### 2.2. Data Sources

The analytical study design used to investigate the causal relationship between AF/L and POAG is presented in Figure 1. The following summary datasets were used: exposure data from the FinnGen genome-wide association study (GWAS) for AF/L, which included 237,690 participants (45,766 cases vs. 191,924 controls), and outcome data from the UKB POAG GWAS, which included 456,351 participants (654 cases vs. 455,697 controls). To assess the causal relationship between AF/L and glaucoma, additional FinnGen data from the following participant types were incorporated in the multivariable MR analysis to assess potential confounders: 377,209 participants with hypertension (111,581 cases vs. 265,626 controls), 281,683 participants with autoimmune hyperthyroidism (1828 cases vs. 279,855 controls), 375,657 participants with sleep apnoea (38,998 cases vs. 336,659 controls) and 377,277 participants with alcohol use disorder (15,715 cases vs. 361,562 controls). Detailed sources of the summary data are presented in Table 1.

### 2.3. Selection of the Genetic IVs

The study used single-nucleotide polymorphisms (SNPs) associated with exposure at the GWAS significance threshold (*p* < 5.0 × $10^{-8}$) as IVs. To maintain the independence of each IV, SNPs were selected after pruning for linkage disequilibrium (*R*^2^ < 0.001, clumping distance = 10,000 kb). Data from the 1000 Genomes Phase III European database were used as a reference to calculate linkage disequilibrium during the clumping process. The *F*-statistic was calculated using the formula
*F* = *R*^2^(*n* − 2)/(1 − *R*^2^)
where *n* represents the sample size, and *R^2^* is the proportion of variance in the exposure explained by the genetic variants [29]. An *F*-statistic >10 suggests no indication of weak instrument bias [30]. For the multivariable MR analysis, the strength of the IVs was evaluated using conditional F-statistics, with values of >10 indicating sufficient instrument strength for the analysis [31].

### 2.4. MR Analysis

In this study, a two-sample MR approach with SNPs was used as IVs to investigate the causal effect of AF/L on POAG. For valid causal inference, IVs were required to meet the following three assumptions: (1) SNPs should be strongly associated with AF/L; (2) SNPs should not be associated with confounders of the AF/L–POAG relationship; and (3) SNPs should influence POAG exclusively through AF/L, with no evidence of directional horizontal pleiotropy. Our primary analytical method was inverse-variance weighting (IVW) with a multiplicative random effects model [30,32,33]. The IVW approach is most efficient when all genetic variations satisfy the three IV assumptions [34]. Recognising the potential bias that may arise from invalid IVs if the assumptions were not fully met, sensitivity analyses were conducted using the weighted median method [35] and MR-Egger regression [36,37]. The weighted median method can provide a consistent estimate even if more than 50% of the IVs are invalid [35]. MR-Egger regression, which is less sensitive to pleiotropy, estimates the average horizontal pleiotropy through its intercept, providing an unbiased estimate of the causal effect [36]. Further, MR-Egger analysis with simulation extrapolation (SIMEX) was employed as it allows bias correction in scenarios where the ‘no measurement error’ assumption is violated, such as when the *I*^2^ value is <90% [37]. To assess heterogeneity and potential pleiotropy, we used Cochran’s Q statistics from the IVW method and Rücker’s Q statistics from the MR-Egger analysis [32,38]. We also employed the MR pleiotropy residual sum and outlier (MR-PRESSO) [39] test and conducted a ‘leave-one-SNP-out’ analysis to identify the influence of potentially pleiotropic SNPs on our estimates. Considering that hypertension, autoimmune hyperthyroidism, sleep apnoea, and alcohol use disorder are common risk factors for POAG, multivariable MR IVW [20] and MR-Egger [40] approaches were used to adjust for these factors and isolate the effects of AF/L on POAG. The Q_A_ statistic, a modification of Cochran’s Q, was used to assess the heterogeneity and potential pleiotropy among IVs [31]. When the conditional *F*-statistic or Q_A_ statistics indicated the presence of weak instruments or potential pleiotropy, the Q-minimisation approach (Q-het) was applied to estimate robust causal associations, supplementing the MVMR-IVW approach. The standard errors were calculated using the bootstrap method [31]. For exposures with overlapping samples, covariances for the effect of each SNP on each exposure were required to calculate the conditional *F*-statistic and Q_A_ statistics [31]. One approach to obtain these covariances is to use a phenotypic correlation matrix [31], which can be derived from the intercept of the bivariate LD score regression [41,42,43]. Causal effects are expressed as odds ratios (ORs) with 95% CIs. Statistical significance was established at a two-tailed *p*-value of <0.05. All MR analyses were performed using the TwoSampleMR, MVMR, Mendelian randomisation, and SIMEX packages in R version 3.6.3 (R Core Team, Vienna, Austria).

## 3. Results

### 3.1. Genetic IVs in Univariable MR

In the MR study, 85 SNPs that were significant at the GWAS level and were independent of each other were selected to serve as IVs. The *F*-statistic values for all the selected SNPs exceeded 10, with a mean value of 114.53, confirming that they were strong instruments. Detailed information on the IVs is provided in Appendix A. According to the results, the assumption of no measurement error was satisfied, with an *I^2^* value of 97.93% (Table 2). Heterogeneity among the IVs was evaluated using Cochran’s Q test from the IVW method and Rücker’s Q’ test from the MR-Egger analysis and found no significant heterogeneity (*p* = 0.338 for Cochran’s Q and *p* = 0.312 for Rücker’s Q’; Table 2). Horizontal pleiotropy was assessed using the MR-PRESSO global test and MR-Egger regression, both with and without SIMEX adjustments; the MR-PRESSO global test indicated no significant pleiotropic effects (*p* = 0.338), and the MR-Egger intercepts showed no evidence of horizontal pleiotropy (intercept = 0.004, *p* = 0.797 without SIMEX; Intercept = 0.004, *p* = 0.848 with SIMEX), as shown in Table 2. Given that the *F*-statistic indicated no evidence of weak instrument bias, the *I*^2^ statistic exceeded 90, and Cochran’s Q test was not significant, the IVW method was selected for primary analysis because of its superior statistical power when core MR assumptions were satisfied [44].

### 3.2. Univariable MR for the Causal Effects of AF/L on POAG

We explored the potential causal relationship between AF/L and POAG by employing 85 SNPs as IVs in a univariable MR analysis. The results, displayed in Figure 2, indicate that AF/L increases the risk of POAG according to the IVW method (OR =1.26, 95% CI = 1.07–1.48, *p* = 0.005). The weighted median approach revealed a similar trend, though it did not achieve statistical significance (OR = 1.25, 95% CI = 0.98–1.60, *p* = 0.068). No significant causal relationship was observed with the MR-Egger method, with or without SIMEX adjustment (OR = 1.20, 95% CI = 0.84–1.74, *p* = 0.323 without SIMEX; OR = 1.22, 95% CI = 0.82–1.81, *p* = 0.329 with SIMEX). Figure 3 displays a scatter plot of the effects of SNPs on AF/L and their corresponding effects on POAG. The leave-one-out analysis corroborated the primary IVW MR findings, as depicted in Figure 4.

### 3.3. Multivariable MR for the Causal Effects of AF/L on POAG

In the multivariable MR analysis (controlled for hypertension, autoimmune hyperthyroidism, sleep apnoea, and alcohol use disorder), the association between AF/L and POAG remained significant (IVW OR = 1.24, 95% CI = 1.02–1.51, *p* = 0.034), as presented in Table 3 (Model 1). Hypertension, autoimmune hyperthyroidism, sleep apnoea, and alcohol use disorder were not significantly associated with POAG in Model 1. The causal effects estimated from the multivariate MR-Egger results were consistent with those of the IVW analysis (Table 3). The Q_A_ statistic for Model 1 was 211.99, with a *p*-value of 0.192, indicating no substantial heterogeneity among the IVs. The conditional *F*-statistics for autoimmune hyperthyroidism, sleep apnoea, and alcohol use disorder were below the conventional threshold of 10 (*F* = 4.08, 3.19, and 3.21, respectively), indicating potentially weak instruments for these exposures. In contrast, the conditional *F*-statistics for AF/L and hypertension were above 10 (*F* = 22.48 and 14.16, respectively), suggesting sufficient instrument strength for these variables. Given the potential bias introduced by weak instruments, we performed an additional multivariate MR analysis (Model 2) that included only AF/L and hypertension, which had conditional *F*-statistics above the threshold of 10. In Model 2, the association between AF/L and POAG remained significant (IVW OR = 1.24, 95% CI = 1.02–1.52, *p* = 0.032), consistent with the findings from Model 1. Hypertension continued to show no significant association with POAG in Model 2 (IVW OR = 0.90, 95% CI = 0.69–1.18, *p* = 0.449). The Q_A_ statistic for Model 2 was 212.86, with a *p*-value of 0.133, indicating no evidence of heterogeneity. To address potentially weak instruments, we also performed an IVW analysis with Q-minimisation (Q-het) for Model 1. The results of this analysis are presented in Appendix A and are consistent with the standard IVW results for AF/L (OR = 1.38, 95% CI = 1.02–2.05).

## 4. Discussion

The findings suggest a potential causal relationship between AF/L and POAG. Moreover, after controlling for the risk factors for AF/L and POAG, including hypertension, autoimmune hyperthyroidism, sleep apnoea, and alcohol use disorders, AF/L demonstrated a causal association with POAG. However, studies on the relationship between AF/L and glaucoma are limited. Potential risk factors for cardiovascular disorders include systemic hypertension, hypotension, vasospasm, elevated blood viscosity, and diabetes, particularly in the absence of elevated IOP [45]. Considering that vascular dysfunction is one of the known mechanisms of glaucoma, AF/L may be considered a cause of glaucoma. According to a longitudinal study on the association between AF and risk of glaucoma [19], AF leads to an increased risk of glaucoma (hazard ratio = 1.31, 95% CI = 1.15–1.48) with variable adjustment. In the sequence of AF onset, conditions are amenable to the formation of an intracardiac thrombus or embolus [46]. According to the literature, AF/L remains one of the main contributors to stroke, unexpected death, and cardiovascular morbidity [13,47]. AF is independently linked to a 1.5-fold higher risk of all-cause mortality in men and a 2-fold higher risk in women [48]. In clinical ophthalmological practice, microembolic complications may remain unreported [49,50]. Reperfusion injury is caused by repeated microemboli occlusion of the central retinal artery, branched posterior ciliary artery, or ophthalmic artery at any level followed by reperfusion, resulting in unstable ocular perfusion of the retina or choroid [18]. Recurrent transient ischaemic episodes can cause retinal ganglion cell death and perfusion disruption, and subsequently glaucoma. Considering that migraine and orthostatic hypotension are risk factors for POAG [51], the presence of AF/L is potentially associated with POAG.

AF/L is characterised by a highly irregular heart rate, which may contribute to unstable ocular perfusion. Given that high IOP is the primary risk factor for POAG, researchers may be curious about the effects of AF/L on IOP as a mechanism of POAG. To address these concerns, the effect of AF/L on IOP through MR analysis indicated that there was no effect on IOP changes (Appendix A). These results support the vascular theory of POAG aetiology, and AF/L may cause POAG because of unstable perfusion pressure rather than IOP. Several studies have suggested a potential association between glaucoma and AF [17]. One study identified correlations between cardiac arrhythmias, particularly AF, with visual field abnormalities and a decline in visual acuity in older patients with glaucoma. AF was also significantly more common in patients with glaucoma than in control participants [52]. Another hospital-based study of Polish patients found that AF, independent of other established cardiovascular risk factors, was associated with an increased risk of normal-tension glaucoma [18]. Normal-tension glaucoma may be influenced by factors other than IOP; however, it is included in the POAG spectrum. Recently, a study using Korean National Health Insurance Service data determined via cross-sectional analysis showed that the AF group had higher incidence rates of diabetes, hypertension, glaucoma, and chronic nephropathy [19]. In addition, a longitudinal analysis showed that patients with AF had a significantly higher cumulative incidence of glaucoma at 11 years (4.37% in the control group vs. 6.42% in the AF group) [19]. In addition, a reduced retinal capillary plexus precedes retinal ganglion cell loss in ocular hypertension [53], and macular microvascular damage is highly associated with visual field defects in POAG [54]. Considering these studies and our results simultaneously, it can be concluded that AF/L and vascular dysregulation can independently cause POAG. Although the primary focus of this study was POAG, angle-closure glaucoma should be considered when interpreting the results. Nevertheless, because this study analysed a large dataset for the risk of glaucoma compared with the severity of AF, its results are expected to be meaningful if interpreted in comparison with the study results.

Hypertension has been suggested as a risk factor for glaucoma, with a higher incidence in patients with POAG (OR = 1.55, *p* < 0.001) [55]. In addition, a Cox regression study found that untreated hypertension led to a greater risk of POAG [56]. AF/L and systemic hypertension frequently coexist with AF because arterial hypertension increases the risk of developing new-onset AF, and because both conditions share common risk factors and underlying pathophysiological mechanisms that contribute to their incidence [57]. A recent MR study indicated that hyperthyroidism is associated with AF, reporting that a genetically elevated FT3:FT4 ratio and hyperthyroidism, rather than FT4 levels within the reference range, are associated with an increased risk of developing AF [58]. Additionally, one study reported that individuals with thyroid eye disease in the All-of-Us Research Program were significantly more likely to be diagnosed with glaucoma [59]. Sleep apnoea is associated with POAG [60], and obstructive sleep apnoea is considered a complex and dynamic substrate for AF [61,62]. Moreover, alcohol consumption is a common trigger for AF/L, especially habitual alcohol intake [63]. A meta-analysis suggested a harmful association between alcohol use and OAG, with IOP elevation [64]. These variables are relevant confounders in studies of the association between AF/L and glaucoma. However, incorporating additional covariates, such as hypertension, autoimmune hyperthyroidism, sleep apnoea, and alcohol use disorder, into our multivariable MR analysis did not alter the significant association between AF/L and POAG. This suggests that the observed relationship between AF/L and POAG was not confounded by these factors, further supporting the causality of the association.

The primary strength of our study lies in the use of a relatively large cohort dataset, which provided evidence suggesting a causal effect of AF/L on POAG. While the study provides robust evidence for a causal relationship between these conditions, it is essential to consider limitations such as the generalisability of the findings to non-European populations, as the dataset was predominantly derived from European cohorts (Finland and the UK). Further research is required to determine whether these results can be generalised to other ethnic groups. Additionally, potential residual confounding factors cannot be ruled out despite multivariable adjustments. This study lacked access to individual-level data, which limited our ability to account for various confounding factors because our analysis relied on summary statistics derived from two-sample MR methods. We were also unable to assess the potential interactions between AF/L and other risk factors, such as hypertension, diabetes, or sleep apnoea, owing to the aggregated nature of the GWAS summary statistics used in our study. Future studies with access to individual-level data are warranted to explore these interactions and their impact on causal pathways. However, the methods employed to validate the MR hypotheses did not provide absolute confirmation. Violations of underlying MR assumptions may lead to erroneous conclusions, necessitating a cautious interpretation of the findings. The conditional *F*-statistics for the confounders included in the multivariable MR analysis were below the conventional threshold of 10 (*F* = 4.08 for autoimmune hyperthyroidism; *F* = 3.19 for sleep apnoea; and *F* = 3.21 for alcohol use disorder), indicating the presence of potentially weak instruments. To address this issue, we performed an IVW analysis with Q-minimisation (Q-het), which is robust against weak instruments and heterogeneity, to validate our findings. Even with the application of robust methods, the non-significant results for certain confounders require cautious interpretation as they may reflect limited statistical power due to weak instruments. Finally, because the prevalence rates of AF/L and POAG were not high (2% and 3%, respectively), GWAS of rare diseases using biobank data may face a severe imbalance in the number of cases and controls [65], which may produce false-positive GWAS hits [66]. When these hits are selected as IVs, they can bias the MR estimates. However, the GWAS summary statistics used in this study employed methods designed to address case-control imbalances, such as SAIGE and the firth test [65,67]. These approaches have been shown to efficiently control false-positive associations and enhance the reliability of selected IVs. Future studies could benefit from the inclusion of larger and more diverse cohorts, as well as the investigation of the specific pathways through which AF/L influences POAG risk.

## 5. Conclusions

The findings contribute to the understanding of the relationship between AF/L and POAG and suggest that AF/L may increase the risk of POAG. This has potential clinical implications as it highlights the importance of monitoring glaucoma risk in patients with AF/L. In addition, the value of this study lies in the recommendation that, in cases where AF/L is diagnosed in a clinical setting, ophthalmological screening should be advised to monitor the development of glaucoma. Conversely, when glaucoma is diagnosed, AF/L should be included in the medical history questionnaire for further evaluation. Further research should explore the biological mechanisms underlying this association and determine whether the effective management of AF/L could reduce the incidence of POAG.

## Figures and Tables

**Figure 1 jcm-13-07670-f001:**
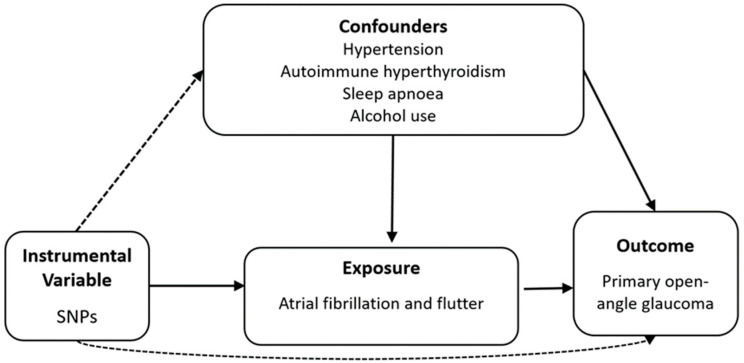
Diagram of two-sample Mendelian randomisation analysis. Solid lines indicate an association, while dashed lines indicate none. Abbreviation: SNP, single-nucleotide polymorphism.

**Figure 2 jcm-13-07670-f002:**
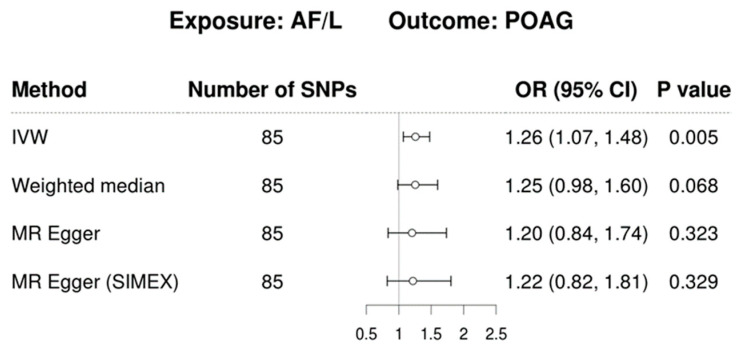
Forest plot of causal association of AF/L with POAG. The plot presents causal effect estimates from various MR methods, including IVW, MR-Egger, and MR-Egger with SIMEX correction. Each method is represented with its corresponding causal estimate and 95% CI. The *x*-axis indicates the OR for POAG associated with the presence of AF/L. Odds ratios greater than 1 suggest that AF/L increases the risk of POAG. Abbreviations: AF/L, atrial fibrillation and flutter; CI, confidence interval; OR, odds ratio; SNP, single-nucleotide polymorphism; IVW, inverse-variance weighting; MR, Mendelian randomisation; POAG, primary open-angle glaucoma; SIMEX, simulation extrapolation.

**Figure 3 jcm-13-07670-f003:**
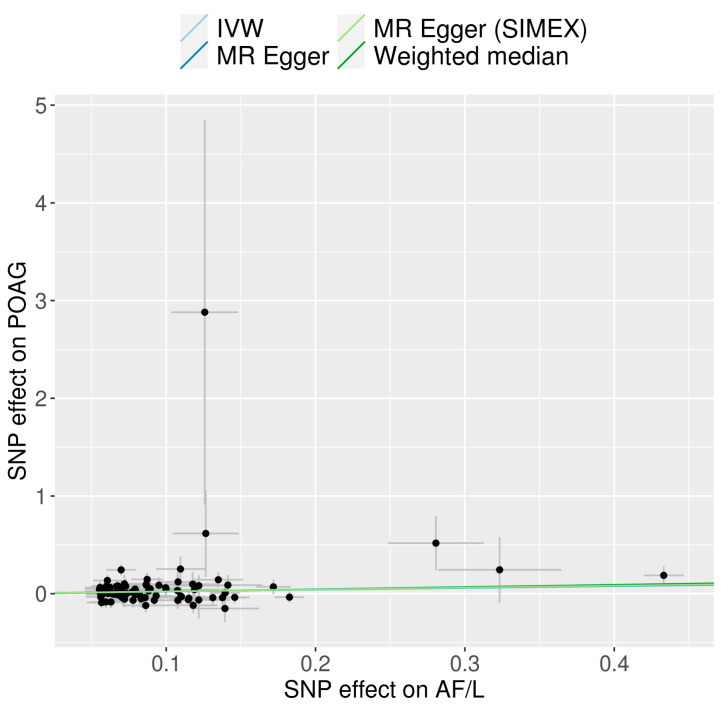
Scatterplot of MR tests showing the effect of AF/L on POAG. Each dot corresponds to an SNP, with the *x*-axis representing the association between the SNP and the exposure and the *y*-axis representing the association between the SNP and the outcome. The regression lines are colour-coded: light blue for IVW, dark blue for MR-Egger, light green for MR-Egger (SIMEX), and dark green for the Weighted Median method. The slope of each line indicates the causal effect estimate obtained through the respective method. Abbreviations: MR, Mendelian randomisation; AF/L, atrial fibrillation and flutter; IVW, inverse-variance weighting; SIMEX, simulation extrapolation; SNP, single-nucleotide polymorphism; POAG, primary open-angle glaucoma.

**Figure 4 jcm-13-07670-f004:**
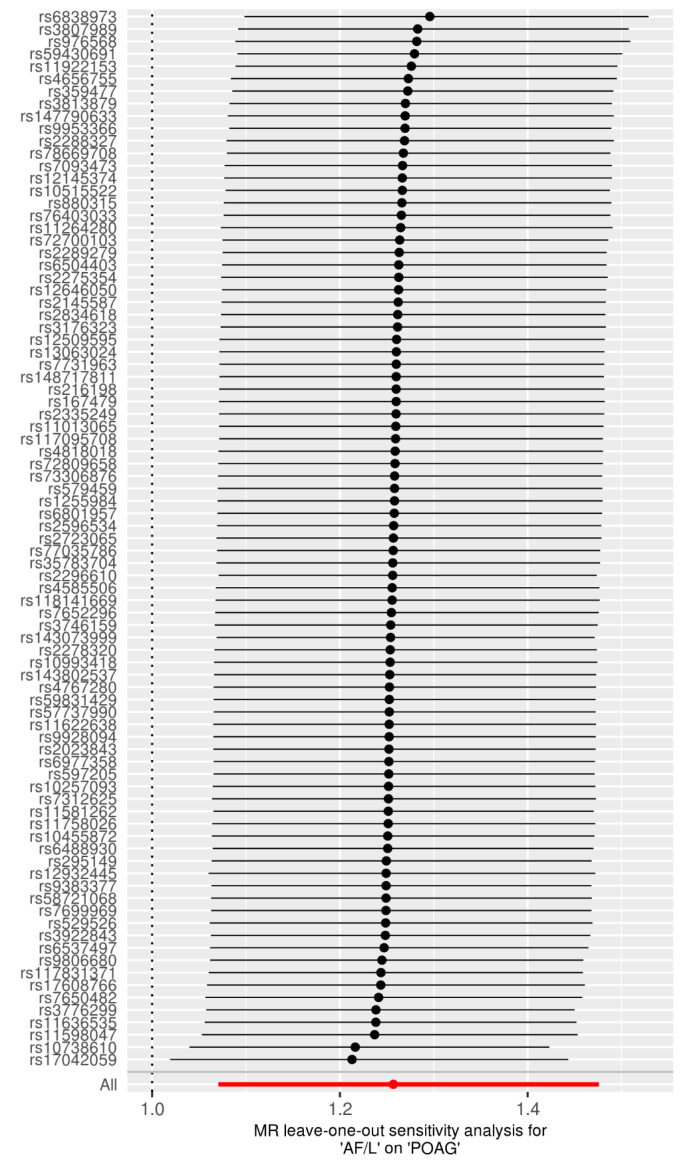
Leave-one-out plot in univariable MR analysis for the effect of AF/L on POAG. Each black dot represents the OR estimate of the causal effect after excluding a specific SNP, with the horizontal line around each dot indicating the corresponding 95% CI. The red dot and horizontal line represent the overall causal estimate (OR and CI) calculated using all SNPs. The *x*-axis represents ORs, where values greater than 1 suggest that AF/L increases the risk of POAG. Abbreviations: MR, Mendelian randomisation; AF/L, atrial fibrillation and flutter; POAG, primary open-angle glaucoma; OR, odds ratio; CI, confidence interval.

**Table 1 jcm-13-07670-t001:** Summary statistics of data sources.

Traits	Data Source	No. of Participants	Population	No. of Variants	Reference
Atrial fibrillation and flutter	FinnGen	237,690 (45,766 cases + 191,924 controls)	European (Finland)	20,164,886	[27]
Hypertension	FinnGen	377,209 (111,581 cases + 265,626 controls)	European (Finland)	20,170,234	
Autoimmune hyperthyroidism	FinnGen	281,683 (1828 cases + 279,855 controls)	European (Finland)	20,167,370	
Sleep apnoea	FinnGen	375,657 (38,998 cases + 336,659 controls)	European (Finland)	20,170,208	
Alcohol use disorder, ICD-based	FinnGen	377,277 (15,715 cases + 361,562 controls)	European (Finland)	20,170,236	
Primary open-angle glaucoma	UKB	456,351 (654 cases + 455,697 controls)	European (UK)	11,831,932	[28]

Abbreviations: ICD, International Classification of Diseases; UK, United Kingdom; UKB, UK Biobank.

**Table 2 jcm-13-07670-t002:** Heterogeneity and horizontal pleiotropy of instrumental variables.

Exposure				Heterogeneity		Horizontal Pleiotropy				
							MR-Egger		MR-Egger (SIMEX)	
	N	*F*	*I*^2^ (%)	*p*-Value *	*p*-Value #	*p*-Value †	Intercept, β (SE)	*p*-Value	Intercept, β (SE)	*p*-Value
Atrial fibrillation and flutter	85	114.53	97.93	0.338	0.312	0.338	0.004 (0.017)	0.797	0.004 (0.018)	0.848

Abbreviations: β, beta coefficient; *F*, mean *F*-statistic; MR, Mendelian randomisation; N, number of instruments; PRESSO, pleiotropy residual sum and outlier; SE, standard error; SIMEX, simulation extrapolation. * Cochran’s Q test from the inverse-variance weighting. # Rücker’s Q’ test from MR-Egger. † MR-PRESSO global test.

**Table 3 jcm-13-07670-t003:** Multivariable MR analysis.

		IVW		MR-Egger	
Exposures	Conditional *F*	OR (95% CI)	*p*-Value	OR (95% CI)	*p*-Value
**Model 1**					
Atrial fibrillation and flutter	22.48	1.24 (1.02, 1.51)	0.034	1.24 (1.02, 1.51)	0.034
Hypertension	14.16	1.00 (0.74, 1.34)	0.984	1.00 (0.74, 1.34)	0.983
Autoimmune hyperthyroidism	4.08	1.03 (0.91, 1.17)	0.617	1.04 (0.90, 1.20)	0.607
Sleep apnoea	3.19	0.60 (0.33, 1.09)	0.092	0.60 (0.33, 1.09)	0.094
Alcohol use disorder, ICD-based	3.21	1.09 (0.80, 1.48)	0.596	1.09 (0.80, 1.48)	0.595
**Model 2**					
Atrial fibrillation and flutter	20.75	1.24 (1.02, 1.52)	0.032	1.28 (1.04, 1.57)	0.019
Hypertension	25.84	0.90 (0.69, 1.18)	0.449	1.12 (0.69, 1.83)	0.637

Abbreviations: MR, Mendelian randomisation; *F*, *F*-statistic; IVW, inverse-variance weighting; OR, odds ratio; CI, confidence interval; ICD, International Classification of Diseases.

## Data Availability

The datasets used and/or analysed in the current study are available from FinnGen (https://finngen.gitbook.io/documentation/data-download, accessed on 4 November 2023) and the GWAS catalogue (https://www.ebi.ac.uk/gwas/summary-statistics, accessed on 4 November 2023).

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
