# Peer review of "Potential Causal Association Between Atrial Fibrillation/Flutter and Primary Open-Angle Glaucoma: A Two-Sample Mendelian Randomisation Study"

_jcm, 2024, doi:10.3390/jcm13247670_

Round 1
Reviewer 1 Report
Comments and Suggestions for Authors
Authors Young Lee, et.al in the article titled “Potential Causal Association between Atrial Fibrillation/Flutter and Primary Open-Angle Glaucoma: A Two-Sample Mendelian Randomisation Study” have investigated to study the relationship between AF/L and POAG may provide clinical insights to the risk factors of atrial fibrillation/flutter (AF/L). Authors have introduced and discussed the topic well and applied satisfactory statistical analysis tools, especially about the application of MR analysis. However, Authors will need to address the following listed comments:
1. - In regard to the clinical significance of this study: Can Authors detail in the conclusion section (section No 5; Line # 276) on cause and effect of the AF/L upon glaucoma and how relevant this study in terms of clinical diagnosis and impact future studies in glaucoma?
2. -Figures 2-4 legends are not clear and will require a well written figure legend to describe the plots for the readers to understand the analysis in brief in relation to the study.
3. - Do Authors have data showing AF/L affecting both the ocular perfusion pressure (OPP) and the Intra ocular pressure (IOP)?
4. -Can Authors provide the rationale for selecting the potential cofounders (hypertension, hyperthyroidism, sleep apnea, and alcohol use disorder) in study, and why it did not show significant association with POAG?
Comments on the Quality of English Language
Typos, English spell check and grammer check is required.
Author Response
Reviewer 1
Authors Young Lee, et.al in the article titled “Potential Causal Association between Atrial Fibrillation/Flutter and Primary Open-Angle Glaucoma: A Two-Sample Mendelian Randomisation Study” have investigated to study the relationship between AF/L and POAG may provide clinical insights to the risk factors of atrial fibrillation/flutter (AF/L). Authors have introduced and discussed the topic well and applied satisfactory statistical analysis tools, especially about the application of MR analysis. However, Authors will need to address the following listed comments:
General Respone) The changes have been marked in red text, and we have provided detailed line numbers to address each point with a point-by-point response.
Comment 1. In regard to the clinical significance of this study: Can Authors detail in the conclusion section (section No 5; Line # 276) on cause and effect of the AF/L upon glaucoma and how relevant this study in terms of clinical diagnosis and impact future studies in glaucoma?
Response 1: Thank you for your valuable feedback and insightful suggestions. We agree that exploring the cause-and-effect relationship between AF/L and POAG is an important aspect that warrants further investigation. Regarding clinical relevance, we have emphasised this more clearly in the conclusion section (lines 355–358)
“In addition, the value of this study lies in the recommendation that, in cases where AF/L is diagnosed in a clinical setting, ophthalmological screening should be advised to monitor the development of glaucoma. Conversely, when glaucoma is diagnosed, AF/L should be included in the medical history questionnaire for further evaluation.”
Comment 2. Figures 2-4 legends are not clear and will require a well written figure legend to describe the plots for the readers to understand the analysis in brief in relation to the study.
Response 2: Thank you for your valuable feedback. We have revamped the legends for Figures 2–4 to offer a more clear and detailed explanation of the plots.
“Figure 2. Forest plot of causal association of AF/L with POAG. The plot presents causal effect estimates from various MR methods, including IVW, MR-Egger, and MR-Egger with SIMEX cor-rection. Each method is represented with its corresponding causal estimate and 95% CI. The x-axis indicates the OR for POAG associated with the presence of AF/L. Odds ratios greater than 1 suggest that AF/L increases the risk of POAG.”
“Figure 3. Scatterplot of MR tests showing the effect of AF/L on POAG. Each dot corresponds to an SNP, with the x-axis representing the association between the SNP and the exposure and the y-axis representing the association between the SNP and the outcome. The regression lines are col-our-coded: light blue for IVW, dark blue for MR-Egger, light green for MR-Egger (SIMEX), and dark green for the Weighted Median method. The slope of each line indicates the causal effect es-timate obtained through the respective method.”
“Figure 4. Leave-one-out plot in univariable MR analysis for the effect of AF/L on POAG. Each black dot represents the OR estimate of the causal effect after excluding a specific SNP, with the horizontal line around each dot indicating the corresponding 95% CI. The red dot and horizontal line represent the overall causal estimate (OR and CI) calculated using all SNPs. The x-axis rep-resents ORs, where values greater than 1 suggest that AF/L increases the risk of POAG.”
Comment 3. Do Authors have data showing AF/L affecting both the ocular perfusion pressure (OPP) and the Intra ocular pressure (IOP)?
Response 3: Thank you for this insightful comment. We agree that investigating the potential effects of AF/L on both OPP and IOP could provide valuable insights. However, after an extensive search for publicly available GWAS summary statistics, we were unable to identify suitable datasets for OPP. IOP data was available for analysis. Therefore, we conducted an additional two-sample MR analysis to assess the causal effect of AF/L on IOP using the available GWAS summary statistics. The results of this analysis have been included in the Supplementary Tables (Supplementary Table 3–5).
(Lines 269–275)
“AF/L is characterised by a highly irregular heart rate, which may contribute to unstable ocular perfusion. Given that high IOP is the primary risk factor for POAG, researchers may be curious about the effects of AF/L on IOP as a mechanism of POAG. To address these concerns, the effect of AF/L on IOP through MR analysis indicated that there was no effect on IOP changes (Supplementary Table 3-5). These results support the vascular theory of POAG aetiology, and AF/L may cause POAG because of unstable perfusion pressure rather than IOP.”
Comment 4. Can Authors provide the rationale for selecting the potential cofounders (hypertension, hyperthyroidism, sleep apnea, and alcohol use disorder) in study, and why it did not show significant association with POAG?
Response 4: Thank you for raising this important question. The potential confounders in our study (hypertension, hyperthyroidism, sleep apnea, and alcohol use disorder) were selected based on evidence from previous research suggesting their association with both AF/L and POAG. These factors share physiological or systemic pathways that could influence both conditions, justifying their inclusion in our multivariable MR analysis. However, although these confounders were pre-specified based on prior knowledge, they were not statistically significant in the final analyses.
In our multivariable MR analysis, only AF/L showed a significant association with POAG. The other confounders did not show a significant association, possibly because of their low conditional F-statistics, indicating weak instrument strength. To address this issue, we included conditional F-statistics in the multivariable MR analysis and used IVW analysis, which is robust to weak instruments. Additionally, we conducted a revised MVMR analysis by excluding confounders with low conditional F-statistics. These additional analyses are expected to clarify the relationship between AF/L and POAG.
We have revised the Methods (lines 143–156, lines 171–174), Results (lines 214–237) and Discussion (lines 333–341) sections to include these points and emphasize the need for cautious interpretation of the non-significant results.
“The QA statistic, a modification of Cochran’s Q, was used to assess the heterogeneity and potential pleiotropy among IVs [31]. When the conditional F-statistic or QA statis-tics indicated the presence of weak instruments or potential pleiotropy, the Q-minimisation approach (Q-het) was applied to estimate robust causal associations, supplementing the MVMR-IVW approach. The standard errors were calculated using the bootstrap method [31]. For exposures with overlapping samples, covariances for the effect of each SNP on each exposure were required to calculate the conditional F-statistic and QA statistics [31]. One approach to obtain these covariances is to use a phenotypic correlation matrix [31], which can be derived from the intercept of the bi-variate LD score regression [41-43]. Causal effects are expressed as odds ratios (ORs) with 95% CIs. Statistical significance was established at a two-tailed P-value of <0.05. All MR analyses were performed using the TwoSampleMR, MVMR, Mendelian randomisa-tion, and SIMEX packages in R version 3.6.3 (R Core Team, Vienna, Austria).”
“Given that the F-statistic indicated no evidence of weak instrument bias, the I² statistic exceeded 90, and Cochran's Q test was not significant, the IVW method was selected for primary analysis because of its superior statistical power when core MR assumptions were satisfied [44].”
“3.3. Multivariable MR for the Causal Effects of AF/L on POAG
In the multivariable MR analysis (controlled for hypertension, autoimmune hyperthyroidism, sleep apnoea, and alcohol use disorder) the association between AF/L and POAG remained significant (IVW OR =1.24, 95% CI =1.02–1.51, P =0.034), as presented in Table 3 (Model 1). Hypertension, autoimmune hyperthyroidism, sleep apnoea, and alcohol use disorder were not significantly associated with POAG in Model 1. The causal effects estimated from the multivariate MR-Egger results were consistent with those of the IVW analysis (Table 3). The QA statistic for Model 1 was 211.99 with a P-value of 0.192, indicating no substantial heterogeneity among the IVs. The conditional F-statistics for autoimmune hyperthyroidism, sleep apnoea, and alcohol use disorder were below the conventional threshold of 10 (F =4.08, 3.19, and 3.21, respectively), indicating potentially weak instruments for these exposures. In contrast, the conditional F-statistics for AF/L and hypertension were above 10 (F =22.48 and 14.16, respectively), suggesting sufficient instrument strength for these variables. Given the potential bias introduced by weak instruments, we performed an additional multivariate MR analysis (Model 2) that included only AF/L and hypertension, which had conditional F-statistics above the threshold of 10. In Model 2, the association between AF/L and POAG remained significant (IVW OR =1.24, 95% CI =1.02–1.52, P =0.032), consistent with the findings from Model 1. Hypertension continued to show no significant association with POAG in Model 2 (IVW OR =0.90, 95% CI =0.69–1.18, P =0.449). The QA statistic for Model 2 was 212.86 with a P-value of 0.133, indicating no evidence of heterogeneity. To address potentially weak instruments, we also performed an IVW analysis with Q-minimisation (Q-het) for Model 1. The results of this analysis are presented in Supplementary Table 2 and are consistent with the standard IVW results for AF/L (OR =1.38, 95% CI =1.02–2.05).”
“The conditional F-statistics for the confounders included in the multivariable MR analysis were below the conventional threshold of 10 (F = 4.08 for autoimmune hyperthyroidism; F = 3.19 for sleep apnoea; and F = 3.21 for alcohol use disorder), indicating the presence of potentially weak instruments. To address this issue, we performed an IVW analysis with Q-minimisation (Q-het), which is robust against weak instruments and heterogeneity, to validate our findings. Even with the application of robust methods, the non-significant results for certain confounders require cautious interpretation as they may reflect limited statistical power due to weak instruments.”
Comments on the Quality of English Language
Comment 5.Typos, English spell check and grammer check is required.
Response 5: Thank you for your comment. To address this concern, we sought an English editing service.
Reviewer 2 Report
Comments and Suggestions for Authors
The current manuscript aims to report potential causal association between atrial fibrillation/flutter and primary open-angle glaucoma. Although the topic is interesting in its scientific field, there are some issues that require the authors’ attention to improve the quality of this particular manuscript.
Specific comments:
1. In Table 1, significant imbalance between the case group and the control group is noted in the data set. The authors should discuss any potential bias arising from this imbalance. Furthermore, the reasonableness of statistical analysis should be justified.
2. Why the inverse variance weighting (IVW) method is chosen as the main analysis? Please clarify.
3. In this report, the data set is mainly from Europe (Finland and the UK). How the findings are generalizable to non-European populations? It is recommended that this limitation should be highlighted in the Discussion section and the studies with a more diverse population should be conducted to validate the results.
4. Are there any interactions between atrial fibrillation/flutter (AF/L) and other risk factors (eg, hypertension, diabetes, or sleep apnea) considered for the analysis? Please specify.
5. In addition to vascular dysregulation, specific biological pathways linking AF/L to POAG should be elaborated. It would be better to include clinical data and relevant mechanisms to give insightful viewpoints.
6. In the Introduction section, the authors mentioned “elevated intraocular pressure (IOP) is a major risk factor for POAG, although the exact pathways of glaucomatous optic neuropathy and the associated visual field loss have not yet been elucidated”. Although their statement is indeed correct, this important scientific claim is not supported by any appropriate documentation. If possible, please consider the inclusion of the following relevant case study (DOI: 10.1016/j.jconrel.2019.11.038) in the reference list to enrich/update article content and attract more attention from broad readers.
Author Response
The current manuscript aims to report potential causal association between atrial fibrillation/flutter and primary open-angle glaucoma. Although the topic is interesting in its scientific field, there are some issues that require the authors’ attention to improve the quality of this particular manuscript.
Respone) Thank you very much for valuable suggestion. The changes have been marked in red text, and we have provided detailed line numbers to address each point with a point-by-point response.
Comment 1. In Table 1, significant imbalance between the case group and the control group is noted in the data set. The authors should discuss any potential bias arising from this imbalance. Furthermore, the reasonableness of statistical analysis should be justified.
Response 1: Thank you for raising this important concern. We acknowledge the significant imbalance between the case and control groups in our dataset. However, our study used a two-sample MR framework, which is less affected by this imbalance than traditional observational studies. GWAS of rare diseases using biobank data can indicate severe imbalance in the number of cases and controls, which may produce false positive GWAS hits. When using such hits as instrumental variables, they may introduce bias in MR estimates. In our study, we used GWAS summary statistics that incorporated methods such as SAIGE and Firth test to address case-control imbalance. These techniques have been proven effective in controlling false positive associations and improving the reliability of the instrumental variables selected. We have revised the Discussion section to include these points and justify the reasonableness of our analysis (lines 342-348).
“GWAS of rare diseases using biobank data may face a severe imbalance in the number of cases and controls [65], which may produce false-positive GWAS hits [66]. When these hits are selected as IVs, they can bias the MR estimates. However, the GWAS summary statistics used in this study employed methods designed to address case-control imbalances, such as SAIGE and the firth test [65,67]. These approaches have been shown to efficiently control false-positive associations and enhance the reli-ability of selected IVs.”
Comment 2. Why the inverse variance weighting (IVW) method is chosen as the main analysis? Please clarify.
Response 2: Thank you for your comments. As suggested, we have revised the text (lines 171-174).
“Given that the F-statistic indicated no evidence of weak instrument bias, the I² statistic exceeded 90, and Cochran's Q test was not significant, the IVW method was selected for primary analysis because of its superior statistical power when core MR assumptions were satisfied [44].”
Comment 3. In this report, the data set is mainly from Europe (Finland and the UK). How the findings are generalizable to non-European populations? It is recommended that this limitation should be highlighted in the Discussion section and the studies with a more diverse population should be conducted to validate the results.
Response 3: Thank you for pointing out the importance of generalizability to non-European populations. We acknowledge this limitation and have revised the Discussion section to highlight that our dataset mainly includes European cohorts from Finland and the UK. We have also added a statement emphasizing the necessity for additional studies in more diverse populations to validate the findings and investigate potential differences in genetic architecture among ethnic groups (lines 319-324).
“While the study provides robust evidence for a causal relationship between these conditions, it is essential to consider limitations, such as the generalisability of the findings to non-European populations, as the dataset was predominantly derived from European cohorts (Finland and the UK). Further research is required to determine whether these results can be generalised to other ethnic groups. Additionally, potential residual confounding factors cannot be ruled out despite multivariable adjustments.”
Comment 4. Are there any interactions between atrial fibrillation/flutter (AF/L) and other risk factors (eg, hypertension, diabetes, or sleep apnea) considered for the analysis? Please specify.
Response 4: Thank you for this insightful comment. We acknowledge the potential importance of interactions between AF/L and other risk factors such as hypertension, diabetes, or sleep apnoea. Our study primarily focused on the causal relationship between AF/L and POAG using an MR approach. However, we were unable to explicitly test for interactions due to the limitations of the GWAS summary statistics used. Two-sample MR relies on aggregated data and does not provide the granularity required to assess interactions. We added this limitation in the discussion section of the manuscript (lines 326-330).
“We were also unable to assess the potential interactions between AF/L and other risk factors, such as hypertension, diabetes, or sleep apnoea, owing to the aggregated nature of the GWAS summary statistics used in our study. Future studies with access to individual-level data are warranted to explore these interactions and their impact on causal pathways.”
Comment 5. In addition to vascular dysregulation, specific biological pathways linking AF/L to POAG should be elaborated. It would be better to include clinical data and relevant mechanisms to give insightful viewpoints.
Response 5: Thank you for this insightful comment. We revised the Discussion accordingly (lines 288-292).
“In addition, a reduced retinal capillary plexus precedes retinal ganglion cell loss in oc-ular hypertension [53], and macular microvascular damage is highly associated with visual field defects in POAG [54]. Considering these studies and our results simultane-ously, it can be concluded that AF/L and vascular dysregulation can independently cause POAG.”
Comment 6. In the Introduction section, the authors mentioned “elevated intraocular pressure (IOP) is a major risk factor for POAG, although the exact pathways of glaucomatous optic neuropathy and the associated visual field loss have not yet been elucidated”. Although their statement is indeed correct, this important scientific claim is not supported by any appropriate documentation. If possible, please consider the inclusion of the following relevant case study (DOI: 10.1016/j.jconrel.2019.11.038) in the reference list to enrich/update article content and attract more attention from broad readers.
Response 6: Thank you for this insightful comment. We have added that distinguished study as a reference (lines 33-38)
“Elevated intraocular pressure (IOP) is a major risk factor for POAG [2], although the precise mechanisms underlying glaucomatous optic neuropathy and the associated visual field loss remain unclear [1]. However, reports indicate that lowering IOP with long-acting drug delivery reduces the risk of glaucoma progression, highlighting that IOP is a critical factor in glaucoma [3].”
Reviewer 3 Report
Comments and Suggestions for Authors
Glaucoma damages not only RGCs and their axons, but the entire visual pathway, including subcortical and cortical centers in the brain.
Tao et al., Pitale et al. published work on a transient increase in IOP in an experiment. In both works, there was a permanent disorder in the capillary plexuses of the retina. So even AF/F can have a similar effect.
Author Response
Glaucoma damages not only RGCs and their axons, but the entire visual pathway, including subcortical and cortical centers in the brain.
Comment 1) Tao et al., Pitale et al. published work on a transient increase in IOP in an experiment. In both works, there was a permanent disorder in the capillary plexuses of the retina. So even AF/F can have a similar effect.
Response 1: Thank you for the positive comments. We added these References in the discussion section (lines 288-292)
“In addition, a reduced retinal capillary plexus precedes retinal ganglion cell loss in ocular hypertension [53], and macular microvascular damage is highly associated with visual field defects in POAG [54]. Considering these studies and our results simultaneously, it can be concluded that AF/L and vascular dysregulation can independently cause POAG.”
Reviewer 4 Report
Comments and Suggestions for Authors
The topic of manuscript is interesting, because authors sought to analyse the association between Atrial flutter and Primary Open-Angle Glaucoma through univariable Mendelian randomisation analysis. To this end authors investigated the causal effects of AF/L on POAG via a two-sample MR approach, using summary statistics from FinnGen for exposures and from the UK Biobank.
Mendelian randomisation is a genetic epidemiological technique that uses genetic variants associated with potential risk factors. The study used single-nucleotide polymorphisms associated with exposure at the GWAS significance threshold (P < 5.0 × 10−8). Were used data from the Genomes Phase III European as the reference to calculate linkage disequilibrium.
The results indicate that AF/L increases the risk of Primary Open-Angle Glaucoma according to the IVW method (p = 0.005). Several studies have suggested a potential link between glaucoma and Atrial flutter. Hypertension is suggested to be a risk factor for glaucoma, with a higher incidence in patients with Primary Open-Angle Glaucoma.
Author Response
Comment 1) The topic of manuscript is interesting, because authors sought to analyse the association between Atrial flutter and Primary Open-Angle Glaucoma through univariable Mendelian randomisation analysis. To this end authors investigated the causal effects of AF/L on POAG via a two-sample MR approach, using summary statistics from FinnGen for exposures and from the UK Biobank.
Mendelian randomisation is a genetic epidemiological technique that uses genetic variants associated with potential risk factors. The study used single-nucleotide polymorphisms associated with exposure at the GWAS significance threshold (P < 5.0 × 10−8). Were used data from the Genomes Phase III European as the reference to calculate linkage disequilibrium.
The results indicate that AF/L increases the risk of Primary Open-Angle Glaucoma according to the IVW method (p = 0.005). Several studies have suggested a potential link between glaucoma and Atrial flutter. Hypertension is suggested to be a risk factor for glaucoma, with a higher incidence in patients with Primary Open-Angle Glaucoma.
Response 1: Thank you for the kind and positive comment.
Round 2
Reviewer 2 Report
Comments and Suggestions for Authors
The revised version has adequately addressed most of the critiques raised by this reviewer and is now suitable for publication in "JCM".